# Harnessing Thalassochemicals: Marine Saponins as Bioactive Agents in Nutraceuticals and Food Technologies

**DOI:** 10.3390/md23060227

**Published:** 2025-05-26

**Authors:** Vicente Domínguez-Arca, Thomas Hellweg, Luis T. Antelo

**Affiliations:** 1Biosystems and Bioprocess Engineering (Bio2Eng) Group, Institute of Marine Research of Spanish Research Council, IIM-CSIC, C/Eduardo Cabello 6, 36208 Vigo, Spain; 2Faculty of Chemistry, Physical and Biophysical Chemistry, Bielefeld University, Universitätsstr. 25, 33615 Bielefeld, Germany; thomas.hellweg@uni-bielefeld.de

**Keywords:** saponins, thalassochemicals, bioactive compounds, nutraceuticals, food technology, marine biomaterials

## Abstract

The expanding field of nutraceuticals and functional food science is increasingly turning to marine-derived bioactive compounds, particularly saponins, for their diverse pharmacological properties. These so-called thalassochemicals display distinctive structural features—such as sulfated glycosidic moieties and amphiphilic backbones—that underpin potent antitumor, hypolipidemic, antioxidant, and antimicrobial activities. In contrast to their terrestrial analogs, marine saponins remain underexplored, and their complexity poses analytical and functional challenges. This review provides a critical and integrative synthesis of recent advances in the structural elucidation, biological function, and technological application of marine saponins. Special emphasis is placed on the unresolved limitations in their isolation, characterization, and structural validation, including coelution of isomers, adduct formation in MS spectra, and lack of orthogonal techniques such as NMR or FTIR. We illustrate these limitations through original MS/MS data and propose experimental workflows to improve compound purity and identification fidelity. In addition to discussing known structure–activity relationships (SARs) and mechanisms of action, we extend the scope by integrating recent developments in computational modeling, including machine learning, molecular descriptors, and quantitative structure–activity relationship (QSAR) models. These tools offer new avenues for predicting saponin bioactivity, despite current limitations in available high-quality datasets. Furthermore, we include a classification and comparison of steroidal and triterpenoid saponins from marine versus terrestrial sources, complemented by detailed chemical schematics. We also address the impact of processing techniques, delivery systems, and bioavailability enhancements using encapsulation and nanocarriers. Finally, this review contextualizes these findings within the regulatory and sustainability frameworks that shape the future of saponin commercialization. By bridging analytical chemistry, computational biology, and food technology, this work establishes a roadmap for the targeted development of marine saponins as next-generation nutraceuticals and functional food ingredients.

## 1. Introduction

In recent years, saponins have gained significant attention across various fields [1,2,3,4,5,6], particularly in nutraceutical research and food technology [1,7,8,9,10]. These naturally occurring glycosides, characterized by their amphiphilic nature, possess a wide range of bioactivities [11,12,13,14,15], making them ideal candidates for human health applications, both as therapeutic agents as well as functional foods and dietary supplements. Their intricate molecular structure, composed of hydrophobic aglycone backbones and hydrophilic sugar moieties, enables them to interact with biological membranes [16,17,18,19], thereby modulating physiological processes. These interactions translate into diverse pharmacological properties, including cholesterol regulation [11,20], immune modulation [21], and cancer prevention [20,22]. All these uses are well documented in both traditional medicine and modern scientific studies. However, the specific molecular mechanisms by which saponins exert their bioactivity—particularly in relation to receptor interactions and signal transduction pathways—are still areas of active research.

This review focuses on the growing potential of marine-derived saponins, particularly from species like sea cucumbers [23] and starfish [15], which possess distinct structural characteristics, such as sulfate groups, that clearly differentiate them from their terrestrial counterparts. These unique features also contribute to the above-mentioned bioactivities and beneficial health uses. While the study of phytochemical saponins [2,21] has greatly advanced our understanding of saponin bioactivity, marine saponins are still underexplored, particularly at the molecular level. Potential mechanisms, such as interactions with lipid bilayers and cellular receptors, remain less characterized, and further research is needed to elucidate their structure–activity relationships. Techniques such as molecular docking [24,25,26,27,28] and machine learning models for molecular parameter and bioactivities predictions [29,30,31,32] could provide valuable insights into these interactions, offering promising directions for future studies.

One of the primary challenges in harnessing the full potential of marine saponins lies in overcoming technical barriers related to extraction [33], stability [34], bioavailability [7], and safety [35]. Traditional extraction methods, such as phase separation-based methods, are widely used [36,37,38], but advanced techniques such as supercritical fluid extraction [39,40], ultrasonic-assisted extraction, and microwave-assisted extraction [33,41,42] are emerging as more sustainable and efficient alternatives. Although saponins typically lack basic amine groups, their amphiphilic structures and the presence of ionizable moieties such as sulfate esters make them susceptible to protonation or altered charge states under acidic conditions. In particular, the low pH of the gastric environment may induce conformational shifts or partial protonation events that influence their solubility, stability, and interaction with biological membranes [7]. This adds further complexity to their pharmacokinetic behavior and analytical characterization.

Despite these physicochemical implications, the literature lacks clear and systematic data on the solubility or self-aggregation behavior of marine saponins. This is largely due to the extreme structural variability found in these compounds—often displaying unusual sugar decorations (e.g., sulfation, acylation, hydroxylation) that increase their hydrophobic character. Additionally, the scarcity of purified marine saponin samples hampers the reproducibility of aggregation assays and physicochemical profiling. As a result, most conclusions about their behavior in aqueous or biological environments remain speculative or extrapolated from terrestrial analogs, underscoring the urgent need for standardized isolation protocols and biophysical testing. Recent advancements in encapsulation and nanotechnology-based delivery systems have shown promise in improving the bioavailability [43,44,45,46] and controlled release of saponins [47], making them more viable for therapeutic and functional food applications.

Furthermore, the increasing demand for sustainable marine resources, making the best and full use of them in a circularity framework, has prompted efforts to exploit marine-derived compounds [48,49], such as saponins, in ways that balance ecological concerns with commercial viability. The sustainable sourcing and production of marine saponins present both opportunities and challenges for the nutraceutical and food technology sectors. Regulatory frameworks and market dynamics also play a crucial role in shaping the incorporation of these compounds into consumer products, influencing both their development and market penetration.

Through this comprehensive review, we aim to shed light on the latest advances in the application of marine saponins in nutraceuticals and functional foods, identifying key challenges and future research directions. By fostering a deeper understanding of the structural, functional, and technological attributes of marine saponins, including the potential integration of computational tools like machine learning for improved molecular parameter estimation, this review seeks to catalyze innovations that will contribute to better customer health outcomes and the sustainable development of saponin-based products.

## 2. Saponin Extraction, Stability, and Bioavailability

Although saponins exihibit promising bioactivities, their industrial use face challenges related to stability and bioavailability, which must be addressed to maximize their therapeutic potential.

### 2.1. Extraction Methods and Challenges

The extraction of saponins from terrestrial and marine organisms presents significant challenges, particularly regarding yield, purity, and environmental sustainability. Traditional methods such as maceration and solvent extraction have been widely used but are often hampered by long processing times, high solvent consumption, and the risk of degrading sensitive bioactive compounds, as well as problems derived from the generation and management of organic solvent wastes or sidestreams. These challenges are particularly acute for marine saponins, where maintaining the integrity of delicate structures, such as sulfated sugar moieties, is essential for preserving bioactivity.

Recent advancements in extraction technologies have addressed some of these limitations. Several of the techniques frequently used can be seen schematized in Figure 1. Supercritical fluid extraction (SFE) [50,51] using supercritical ${\mathrm{CO}}_{2}$, for instance, allows for the extraction of bioactive compounds with high purity and minimal environmental impact. This method is particularly suited for marine saponins, which may be more sensitive to heat and solvents. SFE not only eliminates the need for organic solvents but also preserves the structural integrity of thermolabile compounds. Other techniques, such as ultrasonic-assisted extraction [50,52] (UAE) and microwave-assisted extraction [53] (MAE), improve efficiency by reducing extraction times and increasing yields. UAE uses ultrasonic waves to disrupt cell walls, facilitating the release of saponins, while MAE accelerates solvent penetration and recovery of saponins through microwave radiation.

Advanced methods such as pressurized liquid extraction (PLE) and ultra-high-pressure extraction (UPE) have shown potential in optimizing the extraction of saponins while preserving their structure [50]. Pressurized liquid extraction (PLE), also known as accelerated solvent extraction (ASE), is a technique used to extract compounds from solid or semi-solid samples using solvents at high temperatures (50–200 °C) and pressures (1500–3000 psi). The elevated conditions enhance solvent penetration, reduce extraction time, and increase efficiency by improving solubility and diffusion of saponins. Ultra-high-pressure extraction (UHPE) is a technique that uses extremely high pressures (typically above 100 MPa) to break cell structures and enhance the extraction of bioactive compounds, such as saponins. This method improves yield, reduces solvent use, and preserves heat-sensitive compounds by minimizing thermal degradation, making it a sustainable alternative to conventional extraction techniques.

Optimizing extraction parameters—such as solvent choice, temperature, pressure, and time—is essential for maximizing yield while preserving bioactivity. Additionally, environmental sustainability must be prioritized, particularly in marine ecosystems where overexploitation could have significant ecological impacts. Future research should focus on scaling up these advanced extraction techniques while minimizing their environmental footprint, especially when sourcing marine organisms are used as raw materials to obtain saponins.

### 2.2. Critical Challenges in the Structural Identification of Marine Saponins

Despite the remarkable progress in the characterization of saponins from marine sources, the structural elucidation and purification of these complex glycosides remain a major analytical challenge. This is particularly critical when dealing with triterpenoid or steroidal saponins bearing sulfate groups, commonly present in echinoderms such as holothurians and asteroids.

As highlighted by Decroo et al. [54], saponin extracts typically consist of a complex mixture of congeners, many of them with subtle structural differences that are difficult to separate even by advanced LC-MS protocols. This coelution often prevents the full discrimination of isomeric saponins, especially when tandem mass spectrometry (MS/MS) is performed without prior Q1 isolation or when collision-induced dissociation (CID) fails to produce diagnostic fragments due to low kinetic energy.

Further complications arise from the formation of multiple adducts in electrospray ionization (ESI), including sodium and ammonium adducts, as well as protonated and even diprotonated species [55,56]. These phenomena affect the accuracy of molecular mass determination, the interpretation of fragmentation patterns, and the reproducibility of MS profiles across experiments.

Ion mobility–mass spectrometry (IM-MS) has been introduced as a powerful complementary tool to address this complexity, improving isomer separation and allowing multidimensional analysis of marine saponins [54].

In line with these methodological challenges, we have developed a preliminary study based on extracts from *Cucumaria frondosa*, subjected to reversed-phase HPLC with a C18 column using an acetonitrile gradient (20–100%). Fractionated samples were analyzed via ESI-Q1 in positive mode. In several elution windows, two prominent signals were detected. Fragmentation studies conducted at increasing collision energies revealed characteristic patterns suggestive of saponin-like behavior: prominent fragment ions in the low m/z range (50–200 Da) emerged, which we associate with glycan moieties, while the aglycone fragment proved more resistant to breakdown. This observation is consistent with the known stability of triterpenoid cores under CID [55].

Interestingly, other ion signals from the same fraction displayed distinct fragmentation behavior, with only minor decreases in parent ion intensity upon energy increase, indicative of possible non-saponin contaminants or isomeric saponins with different stabilities.

While a complete structural elucidation (e.g., via NMR) is beyond the scope of this review, these results support the notion that pure isolation of individual marine saponins remains a non-trivial task, and the literature often omits essential MS data (Q1 scans, full fragmentation trees, detailed adduct patterns). We argue that a critical reassessment of current MS-based identification protocols is warranted, and that results based solely on nominal m/z or simplified MS/MS traces should be treated cautiously unless supported by orthogonal techniques such as FTIR, NMR, or ion mobility.

A summary of common pitfalls and suggested analytical strategies is presented in Table 1.

### 2.3. Limitations in Structural Identification and Experimental Proposal

Despite recent advances in LC-MS/MS and molecular networking, the precise identification and structural elucidation of saponins—particularly those from marine sources—remain constrained by several analytical limitations. As evidenced in recent studies [54,55,56], the coelution of isomeric congeners, the formation of multiple adducts (Na^+^, H^+^, etc.), and incomplete MS/MS fragmentation limit confident assignments without complementary NMR or FTIR confirmation. Notably, ion mobility or tandem CID fragmentation often fails to resolve glycosidic linkages or distinguish positional isomers with sufficient accuracy.

Based on these observations, we conducted preliminary experiments using extracts of *Cucumaria frondosa* fractionated by HPLC and analyzed via ESI-Q1 tandem MS under increasing collision energies. Distinct fragmentation profiles suggest the presence of multiple saponins within the same fraction, with some showing a gradual loss of parent ions and glycone fragments and others displaying persistent signals despite increased energy input. These patterns are consistent with the variable stability of glycone and aglycone moieties.

To further illustrate these methodological limitations, Figure 2 presents representative MS and MS/MS spectra obtained from HPLC fractions of *Cucumaria frondosa* extracts. In Figure 2a–c, a saponin-like signal is shown undergoing fragmentation at increasing collision energies (20 and 40 eV), displaying a consistent loss pattern compatible with glycone detachment and suggesting a putative structure. Signals corresponding to singly and doubly protonated species are observed, emphasizing the complexity of ionization states. In Figure 2d–f, another molecular ion exhibits a more complex protonation profile, including [M+H+Na]^2+^ adduct formation. Finally, Figure 2g–i show a signal group with ambiguous fragmentation behavior, where neither a decrease in parent ion intensity nor a defined fragmentation pattern is evident. These observations support the necessity of pure compound isolation and the use of FTIR and NMR as essential orthogonal techniques for confident structural assignment.

Although detailed results will be presented in a separate research article, these findings support the hypothesis that current mass spectrometry workflows are insufficient to achieve complete structural resolution of marine saponins. Therefore, we advocate the incorporation of FTIR and NMR as essential orthogonal methods for confirming saponin identity. This methodological gap, rarely addressed in the literature, may explain discrepancies and redundancies across current saponin databases and justifies a critical reassessment of previously annotated compounds.

### 2.4. Stability and Bioavailability

The stability and bioavailability of saponins are critical factors that determine their efficacy in nutraceuticals and therapeutics. Saponins are prone to degradation under environmental factors such as light, oxygen, and heat, which can significantly reduce their bioactivity [57]. Furthermore, their variable water solubility and limited gastrointestinal absorption present significant challenges for bioavailability.

#### 2.4.1. Enhancing Stability

Various strategies have been employed to enhance the stability of saponins during processing and storage. The use of antioxidants, such as ascorbic acid and tocopherols, can protect saponins from oxidative degradation [58]. Cryopreservation and lyophilization (freeze-drying) have also been adopted to stabilize saponins by removing moisture, preventing both hydrolytic and oxidative degradation. These methods are particularly valuable for saponins intended for pharmaceutical and nutraceutical applications, where maintaining bioactivity over extended periods is essential [59,60].

In addition to antioxidant protection, storing saponins in inert atmospheres (e.g., nitrogen or argon) can minimize oxidation and help preserve their stability. Advances in encapsulation technologies, discussed below, also provide a means to protect saponins from environmental degradation by creating a physical barrier that shields them from harmful light, oxygen, and temperature fluctuations.

#### 2.4.2. Improving Bioavailability: Conflicting Evidence and Future Direction

Improving the bioavailability of saponins has been a major focus of recent research [61,62,63,64]. While many saponins are inherently limited by their poor solubility in water, marine-derived saponins stand out for their notable water solubility. This increased solubility has been attributed, as already mentioned in this work, to the presence of sulfate groups or other hydrophilic moieties, which enhance their bioavailability and play a crucial role in their ecological function as toxins against predators [65]. To this aim, micronization [61,62,66], which reduces particle size, has enhanced solubility by increasing surface area, thereby improving the dissolution rate and absorption of less soluble saponins in the gastrointestinal tract.

Some studies suggest that nanostructured delivery systems offer controlled release [67] and protection from gastrointestinal degradation [68,69,70], although further research is needed to establish their effectiveness for saponin transport specifically. An outline of the strategy for gastroprotection can be seen in Figure 3. Meanwhile, other studies demonstrated complete integrity of the chemical structure of some saponins, and zero metabolization, at least in one species of salmon [71]. It has even been shown that gastric degradation can enhance cytotoxic activity towards cancer cells [72].

However, paradoxically, saponins have been employed as emulsifiers and coating agents in drug delivery formulations designed to facilitate transport through the gastrointestinal tract [73,74,75], as well as transdermal permeation [76]. Some studies have shown that specific saponins enhance the solubility and absorption of co-administered compounds by modulating intestinal permeability and interacting with lipid membranes [77]. This dual functionality raises questions regarding the underlying mechanisms governing saponin stability, transformation, and interaction with biological membranes in different formulations.

The apparent contradiction between saponins’ degradation in the gastrointestinal environment and their role as emulsifiers in drug transport suggests that their stability is highly dependent on their structural characteristics, formulation, and interaction with digestive enzymes and microbiota [78]. Some studies suggest that specific glycosylation patterns or conjugation states determine whether saponins are hydrolyzed into inactive metabolites or act as active transport enhancers [79].

Given these conflicting findings, further research is essential to clarify the precise conditions under which saponins degrade or act as bioavailability enhancers. Systematic studies on different saponin subclasses, their transformation pathways in the digestive system, and their interactions with pharmaceutical carriers will be crucial for optimizing their therapeutic applications and resolving existing paradoxes in the literature.

#### 2.4.3. Recent Technological Advancements

In recent years, several innovations have further improved the extraction, stability, and bioavailability of saponins. Supercritical fluid technology [50,51], initially used for extraction as previously mentioned, has been adapted for encapsulation, providing a solvent-free method for delivering bioactive compounds with minimal degradation. This is especially beneficial for marine saponins, which are sensitive to solvents and thermal stress, as it preserves their bioactivity during the encapsulation process.

Additionally, the use of ionic liquids as alternative solvents has shown promise in saponin extraction. These customizable solvents can be tailored to match the polarity and thermal properties of target saponins, maximizing efficiency while minimizing degradation. This is particularly important for marine saponins, which often require delicate handling due to their complex structures.

Moreover, advances in molecular biology, including CRISPR and synthetic biology, are opening new avenues for improving the intrinsic stability and bioavailability of saponins [80]. Genetic engineering of saponin-producing organisms could lead to novel variants with an increased amount of production, supporting more sustainable production and broader applications in nutraceuticals and pharmaceuticals [81,82,83].

Machine learning (ML) can significantly enhance the bioavailability of saponins by optimizing extraction methods, predicting compound stability, and improving formulation strategies [31,84,85]. By analyzing vast datasets, ML models can identify the most effective delivery systems, such as nanoencapsulation, to enhance solubility and absorption. Additionally, ML-driven molecular modeling can predict interactions between saponins and biological targets, facilitating the design of more bioavailable derivatives [29,30]. These advancements can lead to more efficient utilization of marine resources for pharmaceuticals, nutraceuticals, and cosmetics, maximizing their therapeutic potential.

In conclusion, addressing the challenges of saponin extraction, stability, and bioavailability is crucial for fully realizing their therapeutic and commercial potential. Technological innovations in these areas, particularly in advanced extraction and bioproduction systems, are making saponins more accessible and effective for use in nutraceuticals and pharmaceuticals. Future research should continue to focus on optimizing these processes for large-scale production, while ensuring the sustainability of natural saponin sources, especially from marine environments.

## 3. Physico-Chemical, Molecular, and Biochemical Properties of Saponins

### 3.1. Structure and Classification

Saponins are a diverse group of glycosides characterized by their amphiphilic nature, consisting of a hydrophobic aglycone backbone (sapogenin) linked to one or more hydrophilic sugar chains. This amphiphilicity, with water- and lipid-soluble regions, is essential for their biological functionality, particularly in their interactions with cell membranes. The chemical structure of the aglycone, whether triterpenoid or steroidal, forms the basis for classifying saponins into two main groups. Both types are found in terrestrial and marine organisms. Figure 4 shows a generic characterization of triterpenoid saponins, emphasizing the differences found in both the aglycone moiety and the chemical differences seen in the sugars of marine saponins versus terrestrial ones.

Triterpenoid saponins, predominantly found in plant families such as *Sapindaceae* [86], *Fabaceae* [87], and *Asteraceae* [88,89], display significant structural diversity due to variations in oxidation, glycosylation, and acylation. The oxidation of hydroxymethyl groups in the sugar moieties, often leading to uronic acid formation, enhances solubility and reactivity [90]. These modifications play a crucial role in their interaction with biological membranes and their bioactive properties, contributing to plant defense against pathogens and clearly defining potential pharmacological capabilities, such as anti-inflammatory and cholesterol-lowering activities.

Steroidal saponins, primarily from plant families like *Liliaceae* [91,92] and *Dioscoreaceae* [92,93], share structural similarities with steroid hormones, owing to their 27-carbon skeleton. This resemblance to cholesterol underlies their ability to modulate cholesterol metabolism and absorption, contributing to their hypocholesterolemic [94] and cardioprotective [95] effects.

Marine-derived saponins, such as those from sea cucumbers and starfish, exhibit unique structural characteristics that distinguish them from plant-derived saponins. The incorporation of sulfate groups and rare sugars, such as 4-O-methyl-rhamnose and sulfated fucose, is a hallmark of marine saponins, contributing to their solubility and bioactivity. Sulfation [96,97], in particular, enhances their amphiphilicity and plays a critical role in their potent cytotoxic, immunomodulatory, and antimicrobial properties. For instance, holothurins from sea cucumbers have demonstrated strong cytotoxic effects [98,99], making them candidates for anticancer therapies.

Although saponins are mainly studied in plants and marine organisms, they have also been identified in some other kingdoms, such as fungi [100], where they are thought to serve defensive functions. However, research into fungi-derived saponins is still limited.

In summary, the structural diversity of saponins, driven by factors such as glycosylation, oxidation, and sulfation, directly influences their bioactivity and functionality. Marine saponins stand out due to their unique modifications that enhance solubility and bioactivity, making them valuable for both ecological and biomedical applications.

### 3.2. Biosynthesis and Bioactivities

Saponins are synthesized through complex biochemical pathways, initiated by the cyclization of 2,3-oxidosqualene [101,102]. In plants, this process could follows the mevalonate pathway [103,104], leading to the formation of triterpenoid or steroidal sapogenins. Figure 5 shows a schematic evolution of the chemistry from squalene to triterpenoid or steroid moieties. Oxidation during biosynthesis, particularly in plant saponins, often results in the formation of uronic acids, enhancing solubility and enabling stronger interactions with biological targets, such as cell membranes and pathogens.

In marine organisms, saponin biosynthesis follows similar early steps but diverges significantly in later modifications, with sulfation being a key enzymatic process [105]. This sulfation enhances the amphiphilic properties of marine saponins, allowing them to function effectively in saline environments. Sulfated marine saponins exhibit potent cytotoxic, antimicrobial, and immunomodulatory activities, likely shaped by the harsh conditions of marine ecosystems.

The bioactivity of saponins stems largely from their ability to interact with cell membranes, where they disrupt lipid bilayers [106], increasing membrane permeability and leading to cell lysis in microbial pathogens [107]. In higher organisms, saponins modulate membrane fluidity and influence membrane protein function, contributing to cholesterol-lowering and immune-stimulating effects. In fact, a notable role in stem cell differentiation has also been shown.

The distinct modifications seen in marine saponins, such as sulfation, result in bioactivities that are less common in terrestrial saponins. These differences have significant implications for their therapeutic potential, particularly in cancer treatment, where marine saponins like holothurins have shown promising cytotoxic effects [108]. As research continues, these unique structural modifications may unlock new avenues for biotechnological applications.

In summary, the structural complexity and diverse bioactivities of saponins are closely linked to their biosynthetic pathways and modifications. Marine saponins, with their unique sulfated structures, offer significant advantages in bioactivity and therapeutic potential, particularly in nutraceutical and pharmaceutical development.

### 3.3. Structural Trends and Their Implications for Computational Modeling

The structural complexity of saponins, particularly their glycosylation patterns and amphiphilic nature, directly influences their physicochemical properties and bioactivities. To further explore these relationships, we analyzed a curated dataset of saponins extracted from PubChem (National Center for Biotechnology Information, NCBI), applying systematic filtering criteria to ensure structural consistency. The dataset selection process prioritized molecules featuring triterpenoid or steroidal backbones, including glycosylated derivatives, and was subsequently refined to exclude multi-component entries while retaining those with specific functional moieties, such as sulfate groups or γ-lactone (*O=C1CCCO1*) motifs.

The dataset was constructed by retrieving molecular structures from PubChem that contained at least one of the following core SMILES: *C12CCCC1CCC3C2CCC4C3CCCC4*, *C12CCCC1CCC3C2=CCC4C3CCCC4*, or *C12CCCC1CC=C3C2CCC4C3CCCC4*, ensuring the inclusion of triterpenoid and steroidal scaffolds commonly found in saponins. Entries containing multiple molecules in a single record were excluded unless they only consisted of a primary molecule and a simple counterion (e.g., ${\mathrm{Na}}^{+}$ or ${\mathrm{Ca}}^{2+}$), thus maintaining consistency in molecular representation. Further refinement steps included filtering for molecules containing at least one sugar moiety, and in a more restrictive subset, selecting only those with at least one sulfate or oxofuranose group.

Graphical analysis of key molecular descriptors revealed notable trends among saponins. These trends were initially analyzed for the full dataset (Figure 6), which includes both marine and terrestrial saponins, and subsequently examined in a subset restricted to marine saponins (Figure 7). The comparison of both datasets highlights that the same structural patterns emerge in marine saponins, although with lower data density due to the limited number of marine-specific saponins currently annotated in PubChem.

One of the most evident relationships is the correlation between the number of rotatable bonds (*NER*) and the number of rings (*NA*). Initially, an increase in the number of rings leads to a greater number of flexible linkages, particularly in glycosylated derivatives where sugar moieties contribute to conformational flexibility. However, beyond a certain threshold, this trend stabilizes or even decreases, likely due to the structural constraints imposed by highly interconnected ring systems. In highly fused polycyclic systems, the molecular framework becomes more rigid, limiting additional torsional flexibility.

A similar pattern is observed in the relationship between topological polar surface area (TPSA) and the number of rings. Initially, TPSA increases linearly with ring count, reflecting the addition of polar glycosyl groups. However, at higher ring counts, this trend stabilizes or slightly declines. A plausible explanation is that steric interactions limit the exposure of polar functional groups, reducing the effective polar surface area available for solvent interactions. This behavior suggests that beyond a certain level of glycosylation, additional sugar units may not contribute proportionally to TPSA, potentially affecting solubility and membrane interactions.

From a bioactivity standpoint, TPSA plays a crucial role in determining water solubility, which directly affects both bioavailability and membrane interactions. The observed trend supports the hypothesis that an optimal degree of glycosylation may exist for achieving maximal amphiphilicity, which is critical for the detergent-like behavior of saponins. This property is directly linked to their role in disrupting lipid membranes, a mechanism of action underlying their antimicrobial and immunomodulatory effects.

As mentioned, the overall structural trends are conserved in marine saponins (Figure 7), although the lower number of available structures results in a less pronounced representation.

Given the complexity of these structural relationships, machine learning models represent a promising avenue for predicting saponin bioactivities based on molecular descriptors. By integrating data-driven approaches with structure–activity relationship (SAR) analyses, it may be possible to refine the selection of saponins for specific therapeutic or nutraceutical applications. Future studies should focus on training graph neural networks (GNNs) to recognize structural patterns that correlate with desirable properties, leveraging the trends identified in this dataset as foundational insights for model optimization.

## 4. Structure–Activity Relationships (SARs) of Terrestrial and Marine Saponins

### 4.1. SAR of Terrestrial Saponins

Terrestrial saponins, widely distributed across the plant kingdom, have been extensively investigated for their structure–activity relationships (SARs) due to their therapeutic potential, particularly as anticancer, antifungal, anti-inflammatory, and immunoadjuvant agents.

Recent studies highlight the pivotal role of both the glycosidic chain and the aglycone scaffold in modulating biological activity. For instance, in *Camellia oleifera* saponins, specific positions on the pentacyclic triterpene skeleton—namely C-3, C-15, C-16, C-21, C-22, C-23, and C-28—are crucial for enhancing antitumor, anti-inflammatory, and hypoglycemic effects [109].

In *Aesculus hippocastanum*, esterification at the aglycone level directly influences cytotoxicity. Removal of ester groups dramatically reduces hemolytic activity, underscoring their structural importance [110]. Similarly, studies on escins, isoescins, and deacylated escins reveal that minor stereochemical or regioisomeric differences significantly affect bioactivity, confirming that SAR in these molecules demands fine-grained atomic-level analyses.

The QS-21 saponin from *Quillaja saponaria* exemplifies the impact of targeted modifications. Semisynthetic analogs—e.g., VSA-1 and VSA-2, derived from *Momordica*—incorporating amide replacements at C-28 esters have shown improved stability and reduced toxicity while retaining immunostimulant capacity [111].

Additionally, steroidal saponins from the rhizome of *Anemarrhena asphodeloides* undergo significant chemical transformations upon salt-processing, altering their hypoglycemic profile. Combining plant metabolomics with molecular docking has enabled the identification of bioactive saponins involved in α-glucosidase inhibition [112].

### 4.2. SAR of Marine Saponins

Marine saponins, though less studied, demonstrate exceptional biological potency. OSW-1, a saponin from *Ornithogalum saundersiae*, exhibits cytotoxicity 10–100 times greater than established chemotherapeutics. SAR studies reveal that both the disaccharide moiety and the steroidal aglycone side chain are key to its potency and selectivity. Alterations such as deletion of the C-17 oxo group or disaccharide modifications significantly affect cytotoxicity and calcium-mediated apoptosis [113].

Similarly, Superstolide A, derived from marine sponges, features a 16-membered macrolactone ring and has drawn interest for its synthetic challenges and notable cytotoxicity. Its synthesis has clarified the role of reactions like Julia olefination and Suzuki coupling in assembling such complex molecules [113].

Saponins from sea cucumbers such as pervicoside B and C have also been synthetically accessed to evaluate SAR. Interestingly, simplified analogs retaining the core disaccharide unit preserved antitumor activity, suggesting new leads for future development [114].

### 4.3. QSAR Analysis of Saponins

To complement classical SAR analysis, recent works have applied quantitative structure–activity relationship (QSAR) methodologies—both 2D and 3D—to predict saponin bioactivity across different pharmacological domains.

For instance, QSAR models using multiple linear regression (MLR), partial least squares (PLS), and principal component regression (PCR) demonstrated high predictive accuracy for nematicidal activity of triterpenoid saponins, with *slogP* identified as a crucial descriptor [115]. Similarly, antifungal activity against *Candida albicans* was modeled with high internal ($q^{2}>0.77$) and external ($r^{2}>0.85$) predictivity using k-nearest neighbor molecular field analysis (kNN-MFA) [116].

In silico studies on triterpenoid saponins, such as astragaloside IV, have combined BBB permeability prediction with postmortem brain analysis. Computational descriptors like logBB, ΔlogP, and excess molar refraction (E) demonstrated good correlation with experimental penetration values [117]. Likewise, biomimetic chromatographic techniques combined with QSAR helped predict the ability of saponins such as bacoside A or platycodin D to cross the blood–brain barrier [118].

Advanced QSAR approaches have also been used to model anticancer activity against targets such as KRAS in non-small-cell lung cancer. CoMFA and CoMSIA models based on triterpenoid saponins yielded robust $Q^{2}$ values (>0.77), and ADMET analyses confirmed their pharmacokinetic potential [25].

Finally, QSAR modeling has shed light on the plasma membrane interactions of amphiphilic saponins and other small molecules, emphasizing the relevance of flip-flop dynamics, amphiphilicity, and lipophilicity in targeting strategies [119].

These quantitative models enhance SAR by offering predictive frameworks that integrate physicochemical descriptors and biological activity, allowing for the rational design of more effective saponin derivatives.

### 4.4. SAR of Marine Saponins: Expanded Evidence and Bioassay Correlation

Recent advances have significantly expanded our understanding of marine saponins, especially those derived from sea cucumbers such as *Holothuria scabra*, *Holothuria fuscocinerea*, and *Stichopus herrmanni*. These compounds, generally based on a holostane-type aglycone core and decorated with sulfated oligosaccharide chains, display remarkable antitumor and immunomodulatory activities.

A detailed evaluation of their structure–activity relationships (SARs) reveals that sulfation patterns, the presence of acyl substituents, and specific hydroxylations (such as 17α-OH or 22,25-epoxide) are critical determinants of potency. For example, Holothurin A3 and A4 exhibit potent cytotoxicity against KB and HepG2 cells, with ${\mathrm{IC}}_{50}$ values ranging from 0.32 to 1.12 µg/mL, while Scabrasides A and B show submicromolar ${\mathrm{EC}}_{50}$ values (0.05–0.25 µM) against HL-60 leukemia cells [114,120].

Mechanistic studies point towards membrane disruption and calcium-mediated apoptosis as common pathways, though more targeted assays are needed. Compounds such as Fuscocineroside C and 24-dehydroechinoside A have shown activity in diverse cancer lines, with proposed involvement in mitochondrial destabilization and caspase activation [114].

These findings consolidate marine saponins as promising antitumor agents and validate the continued application of SAR strategies to isolate key pharmacophores and design semisynthetic analogs. A comparative summary of marine saponin structures, activities, and mechanisms is provided in Table 2.

## 5. Saponins in Pharma/Nutraceuticals and Food Technology

### 5.1. Role of Saponins as Nutraceuticals

Saponins, with their diverse structural configurations and broad spectrum of bioactivities, have emerged as valuable compounds in the development of new nutraceuticals and functional foods. Their ability to modulate physiological processes underlies their well-documented hypocholesterolemic, anticarcinogenic, hepatoprotective, hypoglycemic, immunomodulatory, neuroprotective, anti-inflammatory, and antioxidant effects [121,122,123]. These properties also support their integration into formulations for both daily supplementation and targeted therapeutic interventions. Figure 8 shows a synergistic diagram of bioactivities oriented toward biomedicine and bioactivities oriented toward functional food technology. All bioactivities converge on the chemical characteristics of both the aglycone and the glycone identified in marine saponins.

#### 5.1.1. Bioactivities of Saponins

Hypocholesterolemic Activity: Saponins reduce cholesterol absorption in the digestive tract by forming insoluble saponin-cholesterol complexes, which facilitate excretion. This mechanism is well studied in terrestrial sources such as fenugreek (*Trigonella foenum-graecum*) [121] and soybeans (*Glycine max*) [124]. Marine-derived saponins, including asterosaponins from starfish [125], exhibit similar lipid-modulating properties, making them promising candidates for managing hypercholesterolemia and cardiovascular diseases.

Anticarcinogenic Activity: Saponins exhibit anticancer potential by inhibiting tumor growth, inducing apoptosis, and suppressing metastasis through key signaling pathways such as PI3K/AKT and NF-κB [126]. Studies on *Weeping Pittosporum* or *Pittosporum angustifolium* and marine saponins from sea cucumbers highlight their strong cytotoxic effects, supporting further research into their role in cancer prevention and therapy.

Immunomodulatory and Anti-inflammatory Activities: Saponins modulate immune responses by activating antigen-presenting cells, such as dendritic cells and macrophages, enhancing their potential as vaccine adjuvants. *Quillaja saponaria*-derived saponins are widely used in immunostimulating complexes (ISCOMs) [127,128,129] to improve antigen delivery. Additionally, saponins suppress inflammatory pathways by inhibiting cytokines such as TNF-α and IL-6 [130], positioning marine saponins as potential candidates for inflammatory disease treatment and vaccine development.

Hypoglycemic Activity: Saponins contribute to glucose regulation by inhibiting intestinal glucose absorption and modulating hepatic glucose metabolism. Extracts from *Gymnema sylvestre* and marine saponins have demonstrated efficacy in lowering blood glucose levels, making them promising natural agents for hyperglycemia and diabetes management [131].

Antioxidant Activity: By scavenging free radicals and enhancing endogenous antioxidant defenses, saponins help mitigate oxidative stress, a major contributor to chronic conditions such as cardiovascular and neurodegenerative diseases [132]. These properties reinforce their role in disease prevention and overall health maintenance.

#### 5.1.2. Saponins as Nutraceuticals

Leveraging their physiological effects, saponins have been integrated into functional food formulations to enhance health benefits, particularly in cardiovascular, metabolic, and immune-related applications.

Application in Functional Foods: The integration of saponins into functional foods offers a practical approach to leveraging their bioactive properties in daily nutrition [133]. Advances in food technology have enabled their incorporation into diverse products such as beverages, dairy alternatives, cereals, and protein bars. These formulations contribute to cardiovascular support, immune modulation, and inflammation control, aligning with the growing demand for health-promoting dietary options.

Enhancing Bioavailability: A major challenge in saponin utilization is their low bioavailability, primarily due to poor solubility and instability in the gastrointestinal tract [78]. To address this, novel delivery systems such as nanoparticle encapsulation, liposomes, and solid lipid nanoparticles (SLNs) have been developed to protect saponins from enzymatic degradation and improve absorption, ensuring sustained bioactivity and effectiveness [134].

Synergistic Effects with Other Nutraceuticals: Saponins enhance the efficacy of other bioactive compounds when used in combination. Their co-administration with polyphenols, flavonoids, and antioxidants in multi-component formulations has shown synergistic effects, particularly in anti-inflammatory, cardioprotective, and anticancer applications, broadening the scope of nutraceutical innovation [135,136].

Safety and Efficacy Studies: Despite promising preclinical findings, comprehensive clinical trials are crucial for validating saponin health benefits and establishing safety guidelines [137,138]. This is particularly relevant for marine saponins, which may exhibit distinct pharmacokinetics compared to terrestrial saponins. Long-term studies and precise dosage determinations are essential for regulatory approval and consumer acceptance.

Sustainable Sourcing and Biodiversity: The growing demand for saponins necessitates sustainable sourcing strategies. Ethical, sustainable harvesting and controlled cultivation of saponin-producing plants and marine organisms are critical for preserving biodiversity. The exploration of marine biodiversity offers potential for discovering novel saponins with unique bioactivities, while biotechnological approaches could enable sustainable production, supporting global sustainability goals.

In summary, saponins are pivotal in nutraceutical and functional food applications due to their diverse bioactivities. Advances in bioavailability enhancement and sustainability practices will shape the future of saponin-based products, ensuring their continued role in promoting health and disease prevention. Future research should prioritize synergistic formulations, clinical validation, and sustainable sourcing to maximize their impact on health and nutrition.

## 6. Practical Applications and Case Studies

Given their well-established bioactivities, saponins have been extensively studied for their therapeutic potential in clinical settings. Their ability to regulate immune responses, modulate lipid metabolism, and target cancer pathways has positioned them as promising candidates for pharmaceutical applications.

### 6.1. Therapeutic and Biotechnological Uses

Saponins have gained recognition for their diverse therapeutic applications, particularly in vaccine development [111,139,140,141,142,143], oncology [144,145,146], inflammatory diseases [147,148,149,150,151], and the management of cardiovascular [12,152,153] or cerebrovascular [63] diseases. Their immunostimulatory capacity has been leveraged in vaccine formulations, where *Quillaja saponaria*-derived saponins play a crucial role as adjuvants. These compounds enhance antigen presentation by dendritic cells and macrophages, a mechanism effectively utilized in immunostimulating complexes (ISCOMs) designed for vaccines against influenza [154], HIV [155], malaria [156,157], and cytomegalovirus [158]. Their dual ability to boost humoral and cellular immune responses underpins their relevance in next-generation vaccine strategies. Furthermore, they have been successfully used as enhancer factors in human embryonic stem cell culture [159]. Also, some reported results point to strong anti-viral activity of saponins [160,161,162].

In oncology, marine saponins, such as holothurins from sea cucumbers, have shown potential as cytotoxic agents targeting cancer cells while exhibiting selective toxicity. Their role in modulating apoptotic pathways and inhibiting tumor proliferation suggests promising avenues for anticancer drug development. Notably, Holothurin A inhibits prostate cancer growth by reducing PSA expression [163] and modulating androgen receptor (AR) activity through strong binding to the BF3 pocket, as demonstrated by in vitro, in silico, and molecular dynamics studies, suggesting its potential as a treatment for castration-resistant tumors.

Beyond oncology, saponins contribute to the management of chronic inflammatory disorders. Their capacity to modulate cytokine expression, particularly by inhibiting TNF-α and IL-6, offers therapeutic potential in autoimmune conditions characterized by persistent inflammation [164]. These multifaceted properties position saponins as bioactive candidates in both immunotherapeutic and anti-inflammatory drug development.

### 6.2. Cosmetic Innovations

The cosmetic industry has widely embraced saponins for their natural foaming, cleansing, and skin-conditioning properties. These compounds are commonly used in shampoos, body washes, and facial cleansers, where their gentle cleansing action provides a natural alternative to synthetic surfactants. Their ability to form stable foams and improve the sensory qualities of personal care products makes them attractive in formulations designed to meet the increasing consumer preference for natural and sustainable ingredients.

Saponins also exhibit antioxidant and anti-aging properties, making them valuable in modern skincare formulations. By scavenging free radicals and reducing oxidative stress, saponins protect the skin from environmental damage, such as UV radiation and pollution. These properties make them ideal for inclusion in anti-aging creams and serums, where they contribute to skin regeneration, improve elasticity, and reduce the appearance of wrinkles. The multifunctionality of saponins, combining cleansing, antioxidant, and anti-aging properties, positions them as key ingredients in innovative, high-performance skincare products.

Additionally, the antimicrobial properties of saponins enhance their utility in cosmetics by protecting against microbial contamination, improving product safety and shelf life without the need for synthetic preservatives.

### 6.3. Industrial Applications

In the food industry, saponins are highly valued for their emulsifying, foaming, and stabilizing properties. These characteristics make them indispensable in formulating products such as plant-based milks and vegan emulsifiers, where texture and mouthfeel are critical to consumer acceptance. The demand for clean-label, natural ingredients has accelerated the adoption of saponins as natural emulsifiers, replacing synthetic additives in food formulations, such as Tween-80. Their ability to form stable emulsions without affecting sensory properties is a significant advantage in creating consumer-friendly, high-quality products. From an industrial perspective, Figure 9 shows a diagram that structures, based on the bioactivities mentioned in the previous sections, the possible applications related to these bioactivities in the agricultural, cosmetic, or food sectors.

Saponins also extend the shelf life of food products through their antimicrobial and antioxidant properties. By inhibiting microbial growth and delaying lipid oxidation, saponins help preserve the nutritional quality and safety of food products over extended periods. This aligns with the growing market for natural preservatives, where saponins are being incorporated to reduce the reliance on synthetic preservatives, addressing consumer demand for sustainable food products.

In organic agriculture, saponins are used as natural pesticides due to their insecticidal, antifungal, and anti-viral properties. These compounds offer eco-friendly alternatives to synthetic agrochemicals, supporting integrated pest management (IPM) strategies and promoting sustainable farming practices. Saponins have proven effective against a range of agricultural pests, making them valuable in reducing the environmental impact of pesticide use while maintaining crop health.

### 6.4. Research and Development

Ongoing research has driven the development of advanced extraction and purification technologies aimed at improving the yield, purity, and bioactivity of saponin-rich extracts. Techniques such as enzymatic extraction, supercritical fluid extraction, and microwave-assisted extraction have proven effective in isolating saponins with higher purity and reduced environmental impact, minimizing solvent use and energy consumption. These technologies are essential to meet the growing demand for high-quality saponins in both nutraceutical and pharmaceutical industries.

Understanding the critical point of synergy, Figure 10 shows a schematic of the possibilities of chemical junction and the emerging potential. Research has also explored the synergistic effects of saponins with other bioactive compounds, such as polyphenols and flavonoids [45], leading to multi-component nutraceutical products offering enhanced antioxidant and anti-inflammatory benefits. For example, the combination of saponins with polyphenols has been shown to synergistically enhance bioactivity, offering a holistic approach to managing cardiovascular health and reducing inflammation [165]. Such synergistic formulations represent a new frontier in functional foods and nutraceuticals targeting specific health conditions.

Advances in nanotechnology have revolutionized saponin delivery in food and pharmaceutical applications. Encapsulation techniques, such as nanoemulsions, liposomes, and polymeric nanoparticles, have significantly improved the bioavailability of saponins, ensuring more efficient absorption and enhanced therapeutic efficacy [165,166,167,168]. These technologies not only enhance saponin delivery but also provide controlled release mechanisms, prolonging bioactivity and improving consumer outcomes.

### 6.5. The Potential of Thalassochemicals: Marine Saponins

Marine-derived saponins, particularly those from sea cucumbers and starfish, hold significant potential for nutraceutical and pharmaceutical applications due to their unique structures and bioactivities. These marine saponins, characterized by sulfated sugar residues and rare sugar moieties, exhibit a broad range of bioactivities, including cytotoxic, immunomodulatory, and anti-inflammatory effects, making them attractive candidates for health-promoting products.

Holothurinosides, a group of marine saponins, have demonstrated the ability to modulate lipid metabolism by inhibiting pancreatic lipase, thereby reducing fat absorption and promoting weight management [169,170]. These properties make them valuable in the development of nutraceuticals targeting obesity and hyperlipidemia. Furthermore, their potent cytotoxic effects against various cancer cell lines highlight their potential as novel anticancer agents, with studies demonstrating their ability to induce apoptosis and inhibit tumor proliferation in aggressive cancers [163,171,172]. The structural uniqueness of marine saponins, particularly their sulfation patterns, plays a crucial role in enhancing their bioactivity and therapeutic potential.

### 6.6. Conclusion on Practical Applications

Saponins offer a diverse array of practical applications across multiple industries, ranging from therapeutics and food technology to cosmetics and agriculture. Their bioactivities, including immunomodulatory, anticancer, and antioxidant properties, make them invaluable in developing innovative products that cater to the growing demand for natural, sustainable, and health-promoting ingredients.

As research continues to uncover the full potential of saponins, new applications and formulations will undoubtedly emerge. Advances in extraction methods, synergistic interactions with other bioactives, and improvements in bioavailability will shape the future of saponin-based products. In particular, marine saponins represent an exciting frontier in thalassochemicals, offering unique bioactivities that could revolutionize the nutraceutical and pharmaceutical industries.

## 7. Conclusions and Outlook

Saponins have emerged as structurally complex and biologically versatile compounds with growing relevance across the nutraceutical, pharmaceutical, and food technology sectors. While terrestrial saponins have been widely studied, their marine counterparts remain comparatively underexplored, particularly in relation to their sulfated glycosidic chains, molecular stability, and bioavailability. This review has compiled the latest advances in extraction techniques, delivery systems, and computational tools for saponin characterization, highlighting both their potential and the challenges that persist.

The incorporation of saponins into health-promoting products shows strong promise, especially in combating chronic diseases such as cardiovascular disorders, diabetes, and cancer. Their amphiphilic nature allows them to modulate lipid metabolism and immune responses, while their synergistic action with other bioactives such as polyphenols further enhances their therapeutic value.

However, several critical gaps remain. Among them, the accurate structural identification of marine saponins poses a major analytical challenge. Despite the increasing use of advanced mass spectrometry techniques (e.g., ESI-MS/MS, ion mobility, and molecular networking), we find that full Q1 scans, adduct mapping, and comprehensive fragmentation trees are often omitted in the literature. This may lead to misassignments or redundancies in databases and hinders reproducibility. Based on our preliminary experimental evidence with *Cucumaria frondosa*, we advocate for a reassessment of current MS protocols and recommend incorporating orthogonal methods such as NMR and FTIR as standard practices in marine saponin analysis.

Moreover, ensuring bioavailability and structural integrity within biological systems demands ongoing innovation in encapsulation technologies, nanodelivery systems, and synthetic biology strategies. Figure 11 illustrates the challenges that extend beyond extraction as a purely chemical process, highlighting it as a critical and complex step. Regulatory approval continues to depend on thorough clinical validation and safety evaluations, underscoring the need for a clearer pathway from bench to product.

Finally, sustainability must be at the core of future developments, particularly for marine-derived compounds. Ethical sourcing, circular bioeconomy models, and biotechnological production of saponins will be essential to avoid ecological degradation while scaling up for industrial use.

In conclusion, marine saponins represent a promising yet underutilized class of thalassochemicals. Their full potential will only be realized through a concerted effort that bridges analytical chemistry, biotechnology, clinical science, and environmental ethics. With careful stewardship and scientific rigor, saponins can become cornerstone molecules in next-generation functional foods and health applications.

## Figures and Tables

**Figure 1 marinedrugs-23-00227-f001:**
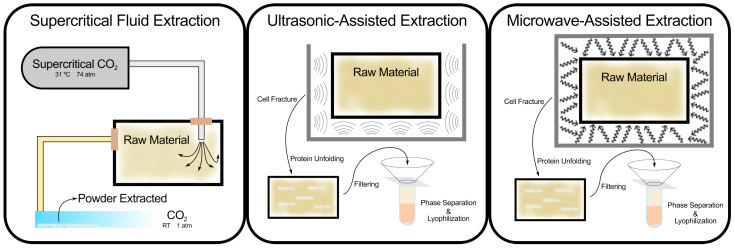
Extraction techniques for saponins, comparing conventional and advanced methods, including supercritical fluid extraction, ultrasonic-assisted extraction, and microwave-assisted extraction.

**Figure 2 marinedrugs-23-00227-f002:**
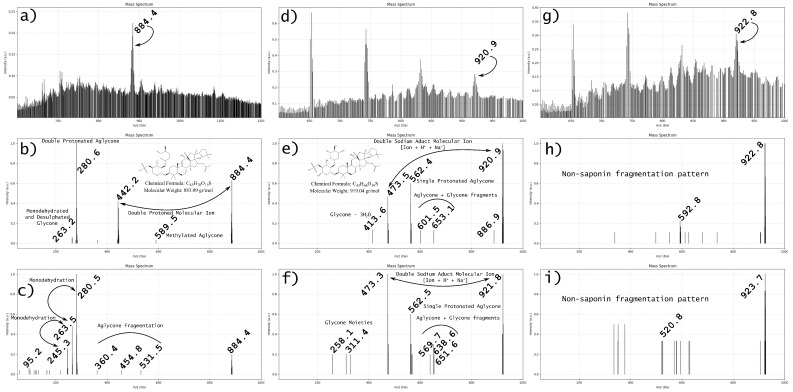
Representative MS and MS/MS spectra from *Cucumaria frondosa* extracts. (**a**–**c**) Q1 and MS/MS at 20 and 40 eV of a presumed saponin showing a clear glycone–aglycone fragmentation pattern; (**d**–**f**) signal showing formation of a [M+H+Na]^2+^ adduct and irregular fragmentation; (**g**–**i**) complex signal group lacking clear fragmentation behavior. These patterns highlight the challenges in compound purity and structural resolution using mass spectrometry alone.

**Figure 3 marinedrugs-23-00227-f003:**
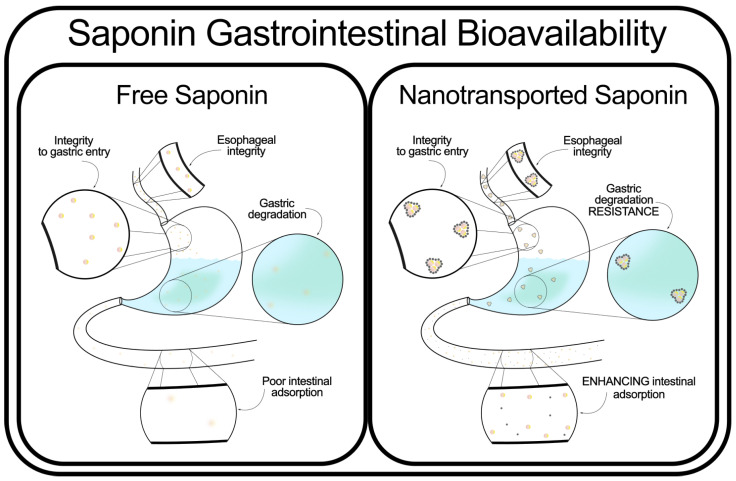
Encapsulation strategies to enhance the stability and bioavailability of saponins, including nanoemulsions, liposomes, and polymeric nanoparticles.

**Figure 4 marinedrugs-23-00227-f004:**
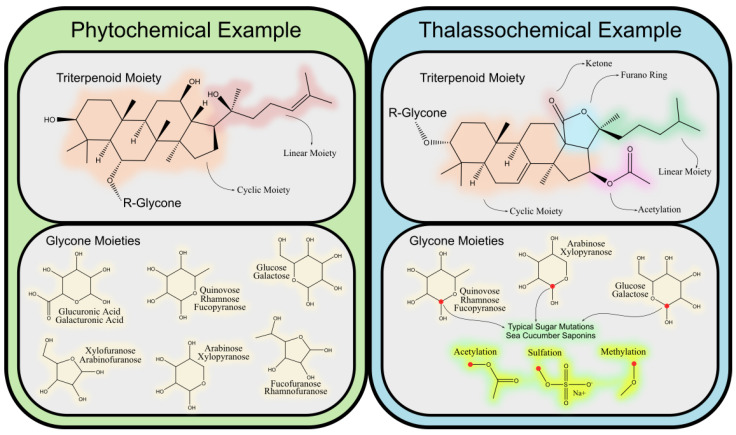
Structural classification of saponins, highlighting key differences between triterpenoid, steroidal, and marine-derived saponins.

**Figure 5 marinedrugs-23-00227-f005:**
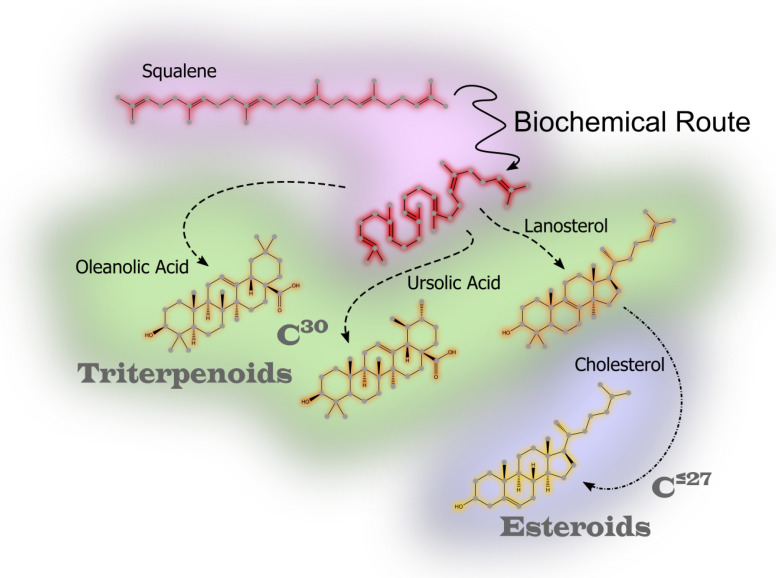
Biosynthetic pathways of saponins in terrestrial and marine organisms, illustrating enzymatic modifications that contribute to their bioactivity.

**Figure 6 marinedrugs-23-00227-f006:**
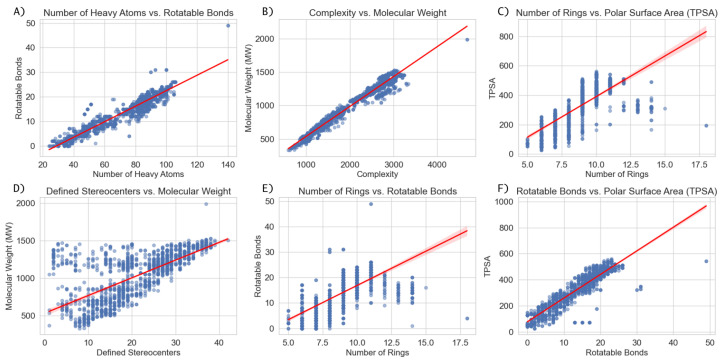
Graphical analysis of molecular descriptors in a curated dataset of saponins, including both terrestrial and marine structures. (**A**) Rotatable bonds vs. number of heavy atoms, (**B**) molecular weight vs. complexity, (**C**) TPSA vs. number of rings, (**D**) molecular weight vs. defined stereocenters, (**E**) rotatable bonds vs. number of rings, and (**F**) TPSA vs. rotatable bonds.

**Figure 7 marinedrugs-23-00227-f007:**
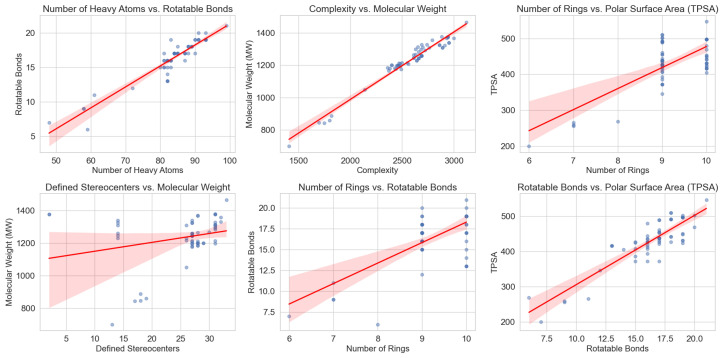
Graphical analysis of molecular descriptors restricted to marine saponins. The same structural trends are observed as in Figure 6, although the lower number of data points results in less pronounced trends.

**Figure 8 marinedrugs-23-00227-f008:**
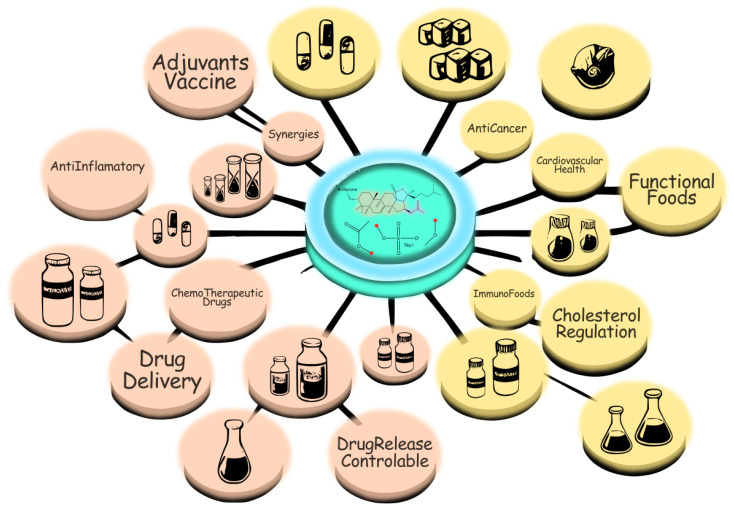
Saponin structures and their bioactivities, illustrating interactions with cellular components and major therapeutic targets.

**Figure 9 marinedrugs-23-00227-f009:**
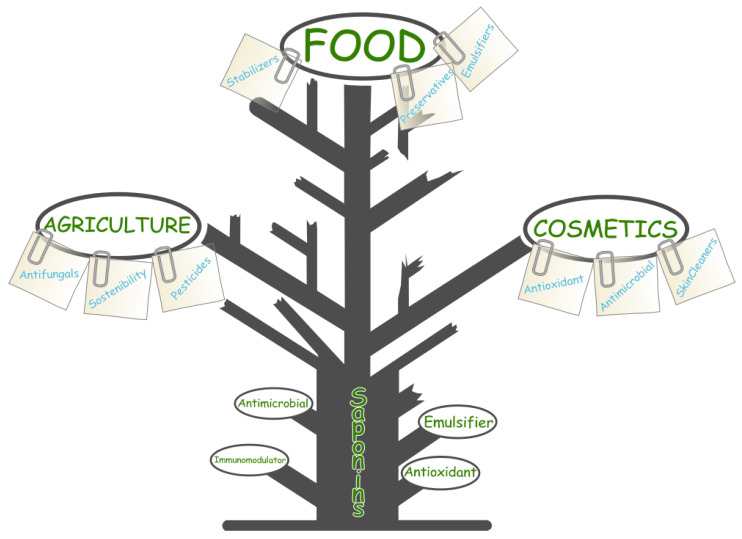
Industrial applications of saponins in pharmaceuticals, functional foods, and cosmetics, emphasizing their role as emulsifiers, adjuvants, and bioactive agents.

**Figure 10 marinedrugs-23-00227-f010:**
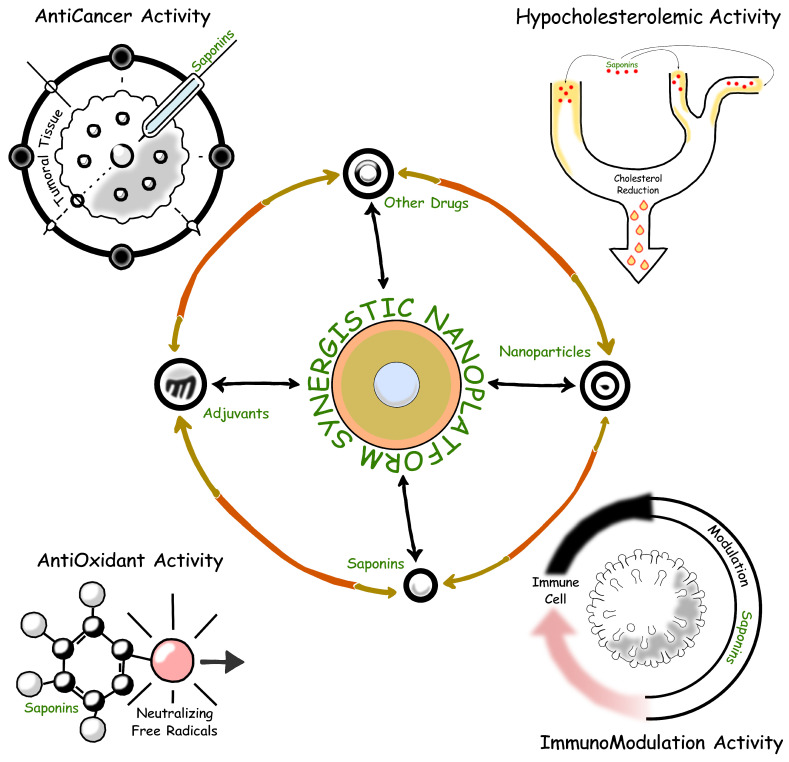
Nanotechnology-based delivery systems for saponins, including nanoemulsions, liposomes, and polymeric nanoparticles, which enhance bioavailability and controlled release.

**Figure 11 marinedrugs-23-00227-f011:**
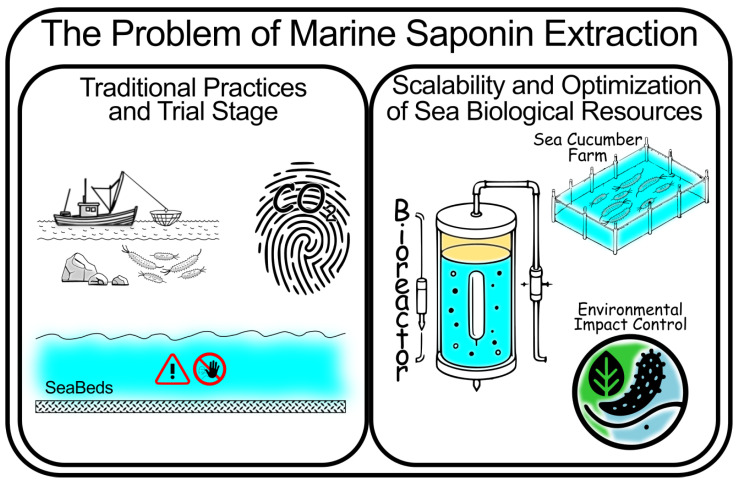
Proposed future directions for marine saponin research, emphasizing sustainability, regulatory challenges, and potential applications in next-generation nutraceuticals.

**Table 1 marinedrugs-23-00227-t001:** Common pitfalls and recommendations in MS-based identification of marine saponins.

Analytical Limitation	Consequence	Recommended Solution
Incomplete Q1 scan before MS/MS	Coeluting ions may be misassigned	Full-range MS1 scan before fragmentation
Overreliance on [M+Na]+ or [M+H]+ adducts	Incorrect MW assignment	Include adduct pattern mapping and isotopic distribution
Lack of collision energy ramping	Missed fragmentation thresholds	Perform stepped CID (e.g., 10, 20, 30, 40 eV)
Absence of NMR confirmation	Misassigned aglycone or glycosidic linkage	Use NMR or at least FTIR if quantity allows
No ion mobility used	Isomeric saponins not separated	Apply IM-MS or alternative orthogonal LC methods

**Table 2 marinedrugs-23-00227-t002:** Structure–activity–mechanism summary of selected marine saponins.

Compound/Saponin	Marine Source	Structural Features	Biological Activity	Mechanism of Action
Holothurin A3	*Holothuria scabra*	Tetrasaccharide chain, holostane-type, sulfate group	Cytotoxic against KB (0.87 µg/mL) and HepG2 (0.32 µg/mL)	Induces apoptosis, membrane disruption
Holothurin A4	*Holothuria scabra*	Tetrasaccharide chain, holostane-type, sulfate group	Cytotoxic against KB (1.12 µg/mL) and HepG2 (0.57 µg/mL)	Induces apoptosis, membrane disruption
Scabraside A	*Holothuria scabra*	Sulfated glycoside, acylated	${\mathrm{EC}}_{50}$ 0.05–0.25 µM (HL-60); moderate on A549	Cell membrane interaction, induces apoptosis
Scabraside B	*Holothuria scabra*	Sulfated glycoside, acylated	${\mathrm{EC}}_{50}$ 0.05–0.25 µM (HL-60); moderate on A549	Cell membrane interaction, induces apoptosis
Fuscocineroside C	*Holothuria fuscocinerea*	22,25-epoxy group, monosulfated tetrasaccharide	${\mathrm{EC}}_{50}$ 0.58 µM (BEL-7402)	Apoptosis via calcium-mediated signaling
Holothurin A	*Holothuria fuscocinerea*	17α-hydroxy, similar to Fuscocineroside C	Less active than Fuscocineroside C (HL-60)	Apoptosis (less potent)
Echinoside A	*Holothuria scabra*	Tetrasaccharide, sulfate, holostane nucleus	Strong cytotoxicity across cancer lines	Apoptosis, possibly via caspase activation
Holothurin B	*Holothuria scabra*	Sulfated glycoside, holostane	Cytotoxic against multiple lines	Likely apoptosis, membrane lysis
24-dehydroechinoside A	*Stichopus herrmanni*	Tetrasaccharide chain, 24-dehydro, holostane core	${\mathrm{IC}}_{50}$ 0.19–1.17 µM across five cancer cell lines	Apoptosis, cell cycle arrest
Holothurin A6	*Stichopus herrmanni*	New glycoside, similar to Holothurin A5	Similar profile to 24-dehydroechinoside A	Similar to 24-dehydroechinoside A (apoptosis)

## Data Availability

The data used for this work will be shared upon request to the authors.

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
