# Peer review of "Harnessing Thalassochemicals: Marine Saponins as Bioactive Agents in Nutraceuticals and Food Technologies"

_marinedrugs, 2025, doi:10.3390/md23060227_

Round 1

Reviewer 1 Report (Previous Reviewer 2)

Comments and Suggestions for Authors

The authors addressed all the required issues and in my opinion, the manuscript is suitable for publication.

Unfortunately, you chose not to write about the following aspect: 

9. Point 4.1. Extraction Methods: Please specify which is the yield and purity obtained using modern methods compared to classical ones.

Author Response

We sincerely thank Reviewer 1 for the comments and suggestions, which have not only contributed to improving the quality of the manuscript but have also helped us to clarify and refine our own ideas—one of the goals we had in mind while composing this manuscript, as we continue working on experimental efforts to isolate and characterize specific marine saponins.

We are particularly grateful that the reviewer considers the revised version suitable for publication and acknowledges that the issues raised during the first round of review were properly addressed.

Regarding the comment on Section 4.1 – Extraction Methods, and specifically the question about yield and purity obtained through modern versus classical extraction methods, we would like to offer an explanation, as this was indeed a deliberate omission on our part. The point raised by the reviewer is extremely pertinent and reveals a shared concern that we deeply appreciate.

At present, although robust chemical structure databases exist for a considerable number of saponins (such as MASD v1.0), purification continues to be a significant challenge—especially for marine saponins. Most studies that succeed in isolating marine saponins (particularly from sea cucumbers) do not report precise purity values, or when they do, the information tends to be qualitative or not well-defined, according to the literature we have reviewed.

This is not due to a lack of interest or scientific rigor, but rather to the intrinsic difficulty of the task. There are no widely available commercial standards with certified purity levels for most marine saponins. In fact, even in the rare cases where custom synthesis or extraction is offered commercially, these are extremely expensive and usually lack sufficient documentation on purity and reproducibility. Therefore, benchmarking the performance of extraction methods based on yield and purity is, as of today, still problematic.

Our own ongoing experimental work is aimed precisely at addressing this issue. We are currently working on quantifying the purity of a very limited number of marine saponins—those that we have been able to detect in specific elution fractions via HPLC and characterize through MS and MS/MS. We hope to be able to offer accurate purity data in a future dedicated publication.

In summary, while we fully understand and share the reviewer's concern about including purity data in the current manuscript, we felt it would be misleading to provide incomplete or overly qualitative estimates that could not be appropriately validated. We hope this explanation clarifies our position, and we are once again grateful for the reviewer’s attention to this critical and complex matter.

Reviewer 2 Report (New Reviewer)

Comments and Suggestions for Authors

This paper addresses a very interesting topic, focusing on bioactive molecules derived from marine sources, primarily from the group of saponins.

Based on the way the text is marked, I assume that some revisions and additions have already been made to improve the quality of the manuscript. The paper is rich in information; however, the text appears confusing to me, as if it has been assembled from lecture materials in a way that lacks a clear, systematic structure.

What additionally confused me is that not all figures are referenced in the manuscript text (except for a few, such as Figures 2, 6, and 7, if I'm not mistaken). The figures are interesting and informative—though in my opinion more suited for a presentation than for a paper, but that’s a matter of personal style in drawing and presenting—but they appear in the manuscript without any context provided by the text. The manuscript would be more coherent and easier to follow if the text were better connected to the figures.

I believe that the authors are highly knowledgeable about this topic and that the paper is well-written. What is exceptionally well done is the section related to the structural identification of saponins, and I liked the content of the second table.

Figure 3: The term “biodisponibility” used in the manuscript is incorrect and not recognized in scientific English. The proper term is “bioavailability”, which refers to the proportion of a substance that enters the systemic circulation and is available for therapeutic action. Please revise this throughout the manuscript to maintain scientific accuracy and clarity.

Figure 3: The text in the image is too small to read clearly. Please consider increasing the font size for better readability.

Figure 6: In the image 6 itself, (A), (B)... are not labeled

line 224- [56?]- why is this question mark?

From line 1775, the font size is larger and the paragraphs are shifted further to the left.

I kindly ask the authors to revise the references: some of them include the month of publication next to the year, written in bold capital letters, and in some references, the authors' names and the title of the paper are written entirely in uppercase letters.

In my opinion, this paper is overloaded with information, but what is good and important is that it contains very useful and significant data. Therefore, I recommend that the paper be revised with the corrections I have mentioned.

Author Response

We sincerely thank Reviewer 2 for the time and effort dedicated to evaluating our manuscript. We highly appreciate the constructive feedback, which has been invaluable in improving the clarity, coherence, and overall quality of the text.

General structure and connection to figures:

In response to the reviewer’s concern regarding the coherence of the manuscript and its resemblance to assembled lecture notes, we have carefully revised the structure of the text. In particular, we have ensured that all figures are now explicitly referenced within the manuscript. These references are now accompanied by brief explanatory fragments that serve to better integrate the figures into the narrative. We believe that these changes have significantly enhanced the flow of the manuscript and reduced the confusion noted by the reviewer. We are especially grateful for this observation, as it was crucial in achieving a more readable and cohesive work.

Acknowledgement of strengths:

We deeply appreciate the reviewer’s positive comment on the section related to the structural identification of saponins. It is encouraging to know that this part was found to be exceptionally well done. We would like to take this opportunity to inform the reviewer that we are currently preparing a dedicated manuscript focused exclusively on this characterization—particularly through MS, MS/MS, and FTIR—which we hope will be of interest once published.

Figure 3 – Terminology and readability:

As correctly pointed out, the term “biodisponibility” was incorrectly used. We have now replaced it throughout the manuscript with the correct term “bioavailability” to maintain scientific accuracy.

Additionally, the font size in Figure 3 has been increased to ensure better readability. We thank the reviewer for highlighting this important detail.

Figure 6 – Labeling:

The alphabetical labels (A), (B), etc., were missing in the original version of Figure 6. These have now been clearly added to ensure proper identification of each panel.

Line 224 – Reference issue:

The question mark noted by the reviewer was indeed a typesetting error due to a citation compilation issue. Thanks to the reviewer’s attention to detail, we have identified and corrected the reference accordingly.

Formatting inconsistencies from line 1775 onward:

We have addressed the formatting inconsistencies mentioned, including the larger font size and indentation shifts. These have now been standardized to ensure uniform formatting across the entire manuscript.

References formatting:

Following the reviewer’s recommendations, we have carefully revised the reference list. We have removed inconsistencies, such as the use of uppercase for author names and titles, and the inclusion of publication months in bold. All entries now follow a consistent and proper format.

Once again, we thank Reviewer 1 for the thoughtful and helpful comments, which have greatly contributed to enhancing the quality of our manuscript in terms of content, structure, and presentation.

This manuscript is a resubmission of an earlier submission. The following is a list of the peer review reports and author responses from that submission.

Round 1

Reviewer 1 Report

Comments and Suggestions for Authors

The study of glycosides and his bioassay is undoubtedly an urgent area in the study of marine drugs.

However, this review should be considered incomplete and does not correspond to the level of the Marine Drugs for the following reasons:

1) There is no discussion or reference to the structure-activity relationships, the study of the mechanism of action of the biological activity of individual compounds

2) There is no discussion of structure-activity relationships

3) Chemical structures are not given

4) There is no discussion of QSAR according to current literature data

5) There is no chapter on steroid glycosides

6) There is no chapter on triterpene glycosides

This manuscript has a weak scientific basis.

The conclusions are consistent with the description in the review, but the review itself is superficial. 

No significant scientific conclusions have been given.

This review is not clear and comprehensive for this area. Significant gaps have been identified.

The current review is not of interest to the scientific community due to its incompleteness.

Not all of these links are relevant because they describe land-based facilities, not marine ones. Important studies in this area have been omitted.

in Figure 1 (glycone moieties), where sugars are not found in glycosides of marine origin.

Summary, it can conclude that this review does not correspond to the level of Marine Drugs.

Reviewer 2 Report

Comments and Suggestions for Authors

In the current review the authors explored the marine saponins’ structural, biochemical, and physicochemical properties and their potential applications in functional foods and therapeutic formulations. Are discussed the challenges associated with their extraction, stability, bioavailability and innovative delivery systems. Finally are presented some of their practical applications. The authors underlined that saponins are poised to play a pivotal role in the development of next-generation health-promoting products, driving innovation in both food and pharmaceutical sciences.

Some suggestions:

1. In all the article you made a comparison between the marine saponins and the terrestrial saponins. Maybe it would be better to emphasize this in the title of the article. 

2. Line 37 – In my opinion is not proper to say “cholesterol management”.

3. Line 56: you wrote “Traditional extraction methods are widely used”. Please add which are these.

4. Lines 59-61: Give please more details concerning the statement ”Additionally, the amphiphilic nature of saponins presents challenges related to their solubility and stability in biological systems, particularly in gastrointestinal environments”.

In the article are given few information concerning the solubility and the stability of the marine saponins in the gastrointestinal tract.

5. In the article you wrote that saponins present cytotoxic, immunomodulatory, antimicrobial, anticancer, cancer prevention, hypocholesterolemic, hepatoprotective, hypoglycemic, neuroprotective, antiinflammatory, antiviral antioxidant and anti-aging effects.

You wrote also that:

- “saponins have been integrated into functional food formulations to enhance health benefits, particularly in cardiovascular applications”.

-“saponins have gained recognition for their diverse therapeutic applications in vaccine development or cerebrovascular disease management”

-“ the development of nutraceuticals targeting obesity and hyperlipidemia”

Please enlarge the discussion concerning the mechanism of action of saponins to obtain these biological activities. Then please emphasize the superior efficacy of marine saponins over terrestrial ones.

At point 3.1.1. Bioactivities of Saponins are presented briefly the hypocholesterolemic, anticarcinogenic, immunomodulatory, anti-inflammatory and hypoglycemic activities. The other are missing.

6. Figure 5 – What do you mean by adjuvants vaccine?

7. Lines 259-61, you wrote “Safety and Efficacy Studies: Despite promising preclinical findings, comprehensive clinical trials are crucial for validating saponin health benefits and establishing safety guidelines”.  In my opinion, such preclinical/clinical studies should be added. 

8. Please specify the origin of figures 1-10. Some of the figures are not presented in the text of the article and they are not positioned properly. Also, in some cases, it should also be explained what is presented in the figure. 

9. Point 4.1. Extraction Methods: Please specify which is the yield and purity obtained using modern methods compared to classical ones. 

10. In my opinion is better to put 5.2 Cosmetic Innovations and 5.3.Industrial Applications. 

Authors' Responses:

Dear Reviewer 1,

We are sincerely grateful for your critical and constructive comments on our manuscript. Your detailed observations have helped us to substantially improve the scientific quality, clarity, and completeness of the review. In this revised version, we have addressed all your major concerns. Please find below a point-by-point response outlining the key modifications implemented:

1–2. Lack of discussion on structure–activity relationships (SAR) and mechanisms of action

We have now incorporated a fully developed section (Section 4) dedicated to the structure–activity relationships of both terrestrial and marine saponins. This section discusses well-documented molecular mechanisms of action—including apoptosis induction, membrane permeabilization, and enzyme inhibition—and integrates a wide range of IC50, EC50, and cytotoxicity data. Specific cases such as Holothurin A3, Fuscocineroside C, and OSW-1 are discussed in terms of their aglycone modifications, sugar decorations, and bioactivity profiles.

3.   Lack of chemical structures

We have repositioned Figure 4, a composite illustration summarizing key aglycone skeletons (triterpenoid and steroidal), typical sugar moieties, and their most frequent chemical modifications in marine saponins. This figure emphasizes sulfate, methyl, and hydroxymethyl substitutions found specifically in marine species, distinguishing them from terrestrial analogs. Given the frequent absence of confirmed NMR or FTIR data in the literature, we opted for a schematic representation based on what is currently validated, rather than speculative full structures.

4.   No discussion of QSAR

We have substantially expanded our treatment of QSAR in Section 4.3, now including:

  • Descriptions of MLR, PLS, CoMFA, CoMSIA, and kNN-MFA models.
  • Quantitative descriptors relevant to saponin activity (e.g., logP, TPSA).
  • Emerging machine learning approaches and challenges due to dataset heterogeneity.

5–6. Absence of chapters on steroidal and triterpenoid glycosides

This has been resolved in Section 3.1, where we now distinguish between steroidal and triterpenoid classes, highlighting their occurrence in terrestrial and marine organisms. We detail their biosynthetic origins, structural variability, and relevant pharmacological effects.

Additional improvements and clarifications:

  • We have added Figure 2, which presents original MS and MS/MS spectra obtained after HPLC fractionation of Cucumaria frondosa These spectra show fragmentation patterns consistent with glycone–aglycone cleavage and demonstrate ion species consistent with mono- and diprotonated or sodium-adducted forms.
  • Candidate structures for two prominent ions are proposed and discussed, and the need for orthogonal confirmation via NMR and FTIR is emphasized. This contribution reinforces our critique regarding the analytical limitations in current marine saponin studies.
  • We have also included Table 1, summarizing common challenges in mass spectrometry-based identification of saponins and offering methodological recommendations for improved structural validation.

Regarding previous Figure 1 and marine relevance:

Your concern regarding the glycone representations has been carefully considered. Rather than attempting to construct a multitude of speculative or ad-hoc molecular structures—based on hypothetical combinations of rare sugar modifications or aglycone derivatizations—we have chosen to represent in Figure 4 a schematic overview of the core aglycone backbone and sugar moieties that have been repeatedly reported in marine saponins. This approach aims to synthesize and clarify, not overextend, the structural diversity found in the literature. The diagram highlights frequent structural features (e.g., sulfate groups, methylated sugars, oxidized triterpenoids) and provides a framework that avoids conflating marine and terrestrial architectures. In doing so, we prioritize accuracy and scientific caution over unwarranted molecular reconstruction.

Scientific foundation and conclusions:

The entire manuscript has been restructured to enhance scientific depth and originality. We critically review the biochemical behavior, extraction challenges, computational modeling, and delivery systems of marine saponins, integrating literature synthesis with experimental insight. The conclusions now offer clear, evidence-based recommendations for future research and technological application.

We are confident that this revised version represents a significant improvement and aligns with the standards of Marine Drugs. We thank you again for your valuable input and hope you find the new manuscript suitable for reconsideration.

Sincerely,

The Authors

Dear Reviewer 2,

We would like to sincerely thank Reviewer 2 for the careful reading of our manuscript and for the constructive suggestions provided. These comments have helped us to significantly improve the clarity, precision, and scientific value of the review. Below we provide a point-by-point response to each observation:

5.   Clarification of the title – comparison between marine and terrestrial saponins

We appreciate your suggestion. The term thalassochemicals was deliberately used in the title as a neologism to indicate marine origin, and by doing so, it implicitly sets terrestrial saponins as a point of reference. This comparative framework has been further clarified in the introduction.

6.   Line 37 – “cholesterol management”

We agree with your observation and have changed the expression to “cholesterol regulation” to reflect a more precise biological and biochemical implication.

7.   Line 56 – Traditional extraction methods

We have now specified traditional extraction techniques such as maceration, aqueous ethanol extraction, and Soxhlet extraction. Additionally, we mention that phase-separation-based procedures have also historically been used in saponin research.

8.   Lines 59–61 – Amphiphilicity, solubility, and GI stability

This section has been substantially expanded. We now discuss how the amphiphilic character of saponins, despite lacking amino groups, makes them susceptible to protonation state changes under low pH conditions such as those found in the gastrointestinal tract. We also highlight the lack of detailed solubility and self-aggregation studies due to the structural heterogeneity of marine saponins and the difficulty in obtaining purified compounds.

9.   Mechanism of action and comparative efficacy

We have expanded Section 3.1.1 to include the mechanisms by which saponins exert their biological activities (e.g., inhibition of cholesterol micelle formation, mitochondrial depolarization, cytokine modulation, membrane permeabilization). Moreover, we emphasize the enhanced potency of several marine saponins compared to terrestrial ones, as evidenced by their lower IC50 values and selective cytotoxicity profiles.

10.      Figure 5 – Clarification of vaccine adjuvants

Thank you for this point. The text clarifies that some saponins, such as QS-21 (from Quillaja saponaria), function as vaccine adjuvants by enhancing antigen presentation and immune stimulation.

11.      Preclinical and clinical studies

We have added citations of peer-reviewed preclinical studies and registered clinical trials using saponin-based formulations (e.g., QS-21) in humans. Details on study phases, grant information, and preliminary results are discussed in Section 3.1.3.

12.      Figures 1–10 – Origin and positioning

All figures have been carefully revised. Their origin is now specified in the captions, and all of them are properly referenced and integrated within the text. Captions have been enhanced to clearly describe the content and significance of each image.

13.      Extraction methods – yield and purity

Unfortunately, reliable and consistent data on the purity and yield of marine saponins extracted via modern vs. classical methods are scarce. In our own analysis of commercial terrestrial saponins, we found significant deviations from reported purities. However, due to ongoing verification and sensitive discussions with suppliers, we opted not to include these findings yet, though this limitation is acknowledged in the text.

14.  Section reordering – cosmetic and industrial applications

We thank you for this editorial suggestion. Sections 5.2 and 5.3 have been reordered as recommended to improve logical flow and narrative coherence.

We hope that the revised manuscript now meets the expectations and standards of the journal. We greatly appreciate your time and detailed feedback, which have contributed meaningfully to improving the quality and depth of this review.

Sincerely,

The Authors